# Reshaping reservoirs with unsupervised Hebbian adaptation

Tanguy Cazalets ⬤ ✉ & Joni Dambre

Reservoir Computing (RC) is a lightweight way to model time-dependent data, yet its reliance on static, randomly initialized network architectures often limits performance on challenging real-world problems. We introduce Hebbian Architecture Generation (HAG), an unsupervised rule that grows connections between neurons that frequently activate together–embodying the biological maxim "neurons that fire together wire together." Starting from an almost empty reservoir, HAG progressively sculpts a task-specific wiring. Across a diverse set of classification and forecasting tasks, reservoirs reshaped by HAG are consistently more accurate than traditional Echo State Networks and reservoirs tuned with popular plasticity rules such as Intrinsic Plasticity or Anti-Oja learning. In other words, letting the network rewire itself from data turns a once-static RC model into a flexible, high-performance learner without a single gradient step. By coupling the efficiency of RC with the adaptability of Hebbian plasticity, HAG moves reservoir computing closer to its biological inspiration and shows that structural self-organization is a practical route to robust, task-aware processing of real-world time-series data.

Reservoir Computing (RC) has emerged as a powerful framework for handling a variety of temporal processing tasks. By transforming input signals into high-dimensional dynamic states through a randomly initialized Recurrent Neural Network (RNN) (the "reservoir"), RC allows complex, nonlinear relationships to be learned with simple linear readouts. However, reliance on a static, random reservoir often leads to suboptimal performance because the network architecture is not tailored to the specific task at hand.

In this paper, we introduce Hebbian Architecture Generation (HAG), an approach that dynamically adjusts the synaptic weights in RNNs to improve their representations of multivariate time-series. HAG is inspired by Hebbian theory[1], the principle that synaptic connections between co-activating neurons strengthen over time, encapsulated by the maxim "neurons that fire together wire together." HAG leverages Hebbian principles to reshape the reservoir based on activity correlations, producing task-specific, high-dimensional feature spaces.

## Static reservoir computing: strengths and limitations

Reservoir computing models that employ discrete-time, rate-based neurons with a continuous activation function are known as Echo State Networks (ESNs)[2,3]. As illustrated in Fig. 1, a typical ESN consists of:

- **Reservoir dynamics** with state vector $\mathbf{x}[t] \in \mathbb{R}^n$ that evolves as

$$\mathbf{x}[t+1] = \sigma\big(\mathbf{W}\mathbf{x}[t] + \mathbf{W_{in}}\mathbf{u}[t] + \mathbf{b}\big) \qquad (1)$$

where $\mathbf{W}$ is the recurrent weight matrix, $\mathbf{W_{in}}$ the $n \times d$ input matrix, $\mathbf{u}[t] \in \mathbb{R}^d$ the $d$-dimensional input, $\mathbf{b}$ a bias vector, and $\sigma$ a nonlinear activation (typically `tanh`). Most ESN studies assume a scalar input; here we treat multivariate time-series, so each channel has its own column in $\mathbf{W_{in}}$

- **Linear readout** that maps the reservoir state to the network output

$$\mathbf{y}[t+1] = \mathbf{W_{out}}\mathbf{x}[t+1] + \mathbf{b_{out}} \qquad (2)$$

where only $\mathbf{W_{out}}$ and $\mathbf{b_{out}}$ are trained using ridge regression.

Reservoirs rely on random projection performed by the reservoir weights into a high-dimensional nonlinear space. This principle is

IDLab, Department of Electronics and Information Systems, Ghent University—IMEC, Gent, Belgium. ✉e-mail: tanguy.cazalets@ugent.be

**K input nodes**  **reservoir with n nodes**  **L output nodes**

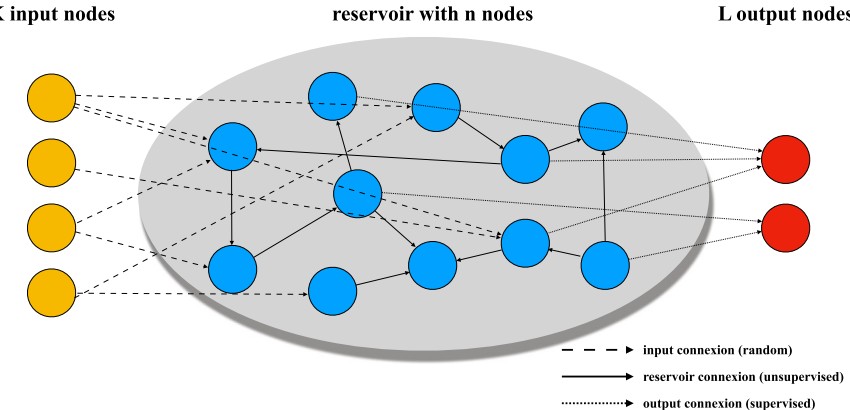

- – – – ▶ input connexion (random)
- ———▶ reservoir connexion (unsupervised)
- ·········▶ output connexion (supervised)

**Fig. 1 | Schematic architecture of an Echo State Network.** A classical ESN consists of an input layer projecting signals into a fixed recurrent "reservoir" and a linear readout. Random recurrent weights transform the input into a high-dimensional dynamic state vector $\mathbf{x}[t]$, from which the readout $\mathbf{y}[t]$ is computed. Only the output weights are trained. The illustration highlights the input weights $\mathbf{W_{in}}$, the internal recurrent matrix $\mathbf{W}$, and the output mapping $\mathbf{W_{out}}$.

supported by Cover's theorem[4], which states that a nonlinear transformation to a higher-dimensional space increases the probability that the transformed patterns are linearly separable.

A key motivation for ESNs is that they avoid the costly (and formerly often unstable, especially before modern practices became widespread) gradient-based training of conventional recurrent neural networks. Because training reduces to a regularized least-squares fit of the output weights, reservoir computing relies on linear optimization with minimal computational resources and can perform well even with relatively small training datasets. ESNs are also widely implemented in hardware "physical reservoir computing" on neuromorphic and photonic substrates, enabling ultra-fast, low-power inference[5–8]. In contrast, gradient-trained RNNs tend to be both data-hungry[9–11] and computationally intensive. This makes ESNs highly competitive with gradient-trained RNNs, especially in low-data or resource-constrained regimes. Empirical studies report that reservoir computers can achieve accuracy comparable to deep learning methods while requiring far fewer samples and training much faster[12–14]. These properties underscore ESNs' suitability for applications where training data or computational budgets are limited.

Although RC has demonstrated success in various sequence modeling applications, limitations remain. The random and static initialization[15] of the reservoir can make performance highly dependent on chance. As ref. 16 describes, this approach is "the antithesis of the optimal" as the reservoir remains unchanged regardless of task requirements. Additional challenges highlighted by ref. 17 include the absence of unsupervised adaptation, unclear criteria for reservoir suitability, and limited biological plausibility.

**Previous work**

Several approaches have attempted to introduce neuroplasticity to enhance performance and adaptability in RC.

Ref. 3 provides an overview of various early work on unsupervised methods that have been used to improve RC. Among early efforts to improve ESN performance through plasticity,[18] achieved good results by integrating an Intrinsic Plasticity (IP) rule tailored to recurrent architectures. A particularly notable approach was proposed by ref. 19, who introduced an Anti-Oja's rule[20] to improve the prediction accuracy of ESNs on chaotic time-series, such as the Mackey-Glass system.

Subsequent research explored various combinations of plasticity rules to optimize reservoir performance. For example, in early work,[21] investigated structural adaptation of binary (on/off) neural network by blending spike-inspired rules (allowed by the binary nature of the network they investigated) on a specific benchmark of their making. refs. 22 and [23] produced more generalizable work with tanh and

sigmoid activation functions by testing performances of Anti-Oja's learning with intrinsic plasticity (IP) and combination of the two, achieving incremental performance improvements for time-series forecasting. Additionally, refs. 24 and [25] explored heterogeneous applications of IP, Oja, and Anti-Oja rules, where synaptic plasticity parameters were different for each neuron of the network finding that this diversity slightly improved performance. The BCM rule[26] has also been examined by ref. 27 but the authors reported limited and poorly reproducible results, while[28] recently introduced a novel BCM rule tailored to delay-sensitive networks. Other investigations include[29], who demonstrated that a simple homeostatic-based rule with Hebbian effects could marginally improve performance.

Parallel lines of research have examined task-adaptive RC using physical substrates rather than traditional simulated networks. ref. 30 presented a physical RC framework that adapts its computational properties such as nonlinearity and memory, tuned via phase changes. In the domain of spiking neural networks, studies such as refs. 31 and [32] achieved performance improvements over static Liquid State Machines (LSMs)[33] by incorporating BCM, STDP, and TP-STDP rules.

This body of work collectively highlights the ongoing efforts to move beyond static, random reservoir architectures toward more biologically inspired, adaptive, and task-specific networks.

**Contributions**

Like prior efforts, HAG attempts to overcome key RC limitations by addressing: (1) Unsupervised, task-specific adaptability: By forming synaptic connections dynamically, HAG tailors reservoir structure to the task at hand. (2) Biological insights: Reflecting the principles of Hebbian and structural plasticity, HAG mimics the adaptability of biological neural networks by reorganizing connectivity to optimize input feature combination.

However, the core idea behind HAG (growing reservoir connectivity based on neuron activations over extended time windows) distinguishes it from prior plasticity-based techniques in several key ways:

1. Structural plasticity from scratch: Unlike approaches that fine-tune existing connectivity, HAG begins with an empty connectivity matrix and gradually builds synaptic connections between frequently co-activating neurons. This design is inspired by evidence that biological circuits reorganize through Hebbian-like processes of synaptic growth and retraction[34–36].
2. Longer scale linear correlation as a driver: Prior plastic ESNs typically operate on a moment-to-moment basis: if two neurons' activations co-occur, the synaptic weight between them is updated immediately. By contrast, our method explicitly

computes the linear correlation coefficient over a longer time window, thereby capturing a more global or statistical view of neuronal co-activation. This extended perspective yields more robust and informative connectivity updates.

3. Task-specific adaptation to genuinely multivariate streams: Prior plasticity-based ESNs are often demonstrated on scalar or low-dimensional signals and handle channels largely in isolation. HAG targets high-dimensional, highly correlated time-series by wiring together units whose activities co-fluctuate in a statistically robust way across the full state vector. This selectively amplifies cross-channel structure that improves downstream separability and suppresses redundant couplings, i.e., it learns a *task-relevant subspace* instead of blindly increasing the dimensionality.

4. More ambitious task scope: While much of the adaptive ESN literature focuses on forecasting tasks, HAG demonstrates significant gains in classification as well. Its emphasis on data separability and unsupervised structural adaptation broadens the applicability of adaptive RC methods to a wider range of problems.

In the following, Section Results presents empirical results showing improved performance of HAG over traditional RC and plastic RC across multiple benchmarks/tasks and we analyze how HAG's correlation-based restructuring translates into different embeddings. Section Methods introduces the HAG algorithm in detail and outlines the experimental setup.

## Results

We report test performance using the best cross-validated hyperparameters for each model variant. Tables 1 and 2 present (i) classification accuracy and (ii) 5-step-ahead forecasting NRMSE both averaged over 8 independent trials with different seeds.

For classification, HAG variants remain among the top performers on most tasks, robustly outperforming all ESN architectures evaluated,

mean-HAG seems to achieve best on the smallest datasets and variance-HAG takes the lead on bigger datasets. Both HAG variants take the overall lead on the smaller datasets (Japanese Vowels, Cats-Dogs); HAG remains competitive over gradient based models on the medium-sized datasets (FSDD and Spoken Arabic Digits); results on larger dataset (Speech Commands) are dominated by gradient based models. This illustrates the trade-off between fully trainable recurrent networks and reservoir computing (GRU and LSTM) and reservoir computing, with the former achieving the highest accuracies, highlighting the advantage of gradient-based training when abundant data are available (See Table 3 for datasets sizes). However, this comes at the cost of substantially longer training times and greater risk of overfitting in low-data regimes. The rightmost column reports the mean rank of each model across the five classification tasks (lower is better). HAG variants achieve the lowest average ranks, while GRU and LSTM follow closely behind.

Table 2 reports the NRMSE for 5 step ahead forecasting, averaged over eight trials. Here, too, the two HAG variants provide competitive or superior performance on the Mackey-Glass dataset, but on the Lorenz and Sunspot datasets the gradient trained GRU and LSTM models achieve significantly lower errors. This reflects the known strength of gradient trained recurrent networks in modeling smooth, continuous dynamics when sufficient training data are available. Conversely, the poor performance of the GRU and LSTM on Mackey-Glass underlines their sensitivity to training data quantity and the advantage of closed form training in RC. The rightmost column lists the average rank across the three forecasting tasks; lower ranks indicate better overall performance. Mean-HAG remains above static reservoirs and variance-HAG comparable to local rule ESNs.

In time-series forecasting tasks, both HAG variants surpass most baselines but offer only marginal improvements. Our results contrast with the studies by refs. 22 and [23] by showing that in most instances IP + Anti-Oja does not necessarily outperform static or other plasticity rule reservoirs, while performing worse than HAG.

## Table 1 | Test classification accuracy (mean ± s.d.) over 8 trials; values are in %

|  | Japanese Vowels | CatsDogs | FSDD | Spoken Arabic Digits | Speech Commands | Cumulative rank |
|---|---|---|---|---|---|---|
| **E-ESN** | 97.3 ± 0.4 | 67.2 ± 0.6 | 17.6 ± 0.8 | 71.6 ± 1.7 | 6.0 ± 0.1 | 34 |
| **ESN** | 97.5 ± 0.7 | 64.5 ± 1.3 | 24.3 ± 2.5 | 85.9 ± 1.3 | 6.6 ± 0.8 | 33 |
| **IP** | 96.9 ± 0.6 | 66.7 ± 2.3 | 22.1 ± 2.6 | 88.5 ± 1.2 | 10.7 ± 1.1 | 28 |
| **Anti-Oja** | 96.8 ± 0.3 | 66.8 ± 1.6 | 26.2 ± 1.7 | 86.3 ± 0.9 | 8.0 ± 1.0 | 30 |
| **IP + Anti-Oja** | 96.9 ± 0.4 | 65.6 ± 0.7 | 22.7 ± 1.5 | 87.3 ± 1.1 | 10.1 ± 0.9 | 30 |
| **LSTM** | 85.6 ± 3.1 | 63.9 ± 6.9 | **51.7 ± 2.9** | <u>97.7 ± 0.7</u> | <u>77.9 ± 12.8</u> | 23 |
| **GRU** | 93.3 ± 1.6 | 64.8 ± 2.7 | 45.4 ± 18.9 | **98.5 ± 0.4** | **89.9 ± 0.3** | 21 |
| **Mean-HAG** | **98.8 ± 0.1** | **68.8 ± 0.7** | 47.4 ± 2.1 | 96.0 ± 0.4 | 11.1 ± 4.1 | **12** |
| **Variance-HAG** | <u>98.4 ± 0.3</u> | <u>67.9 ± 1.1</u> | <u>48.0 ± 0.8</u> | 95.9 ± 0.4 | 30.8 ± 0.5 | <u>13</u> |

Cumulative rank is unitless. Bold indicates the highest mean per dataset; underlined values are the second-best.

## Table 2 | NRMSE (mean ± s.d.) for a 5-step-ahead forecast evaluated over 1000 time steps (averaged over 8 trials)

|  | Mackey-Glass | Lorenz | Sunspot | Cumulative rank |
|---|---|---|---|---|
| **E-ESN** | **0.00319 ± 0.00021** | 0.788 ± 0.00184 | 0.314 ± 0.00202 | 15 |
| **ESN** | 0.00454 ± 0.00037 | 0.809 ± 0.00531 | 0.313 ± 0.00078 | 17 |
| **IP** | 0.00442 ± 0.00022 | 0.810 ± 0.01135 | 0.312 ± 0.00153 | 16 |
| **Anti-Oja** | 0.00473 ± 0.00031 | 0.809 ± 0.00208 | 0.313 ± 0.00153 | 19 |
| **IP + Anti-Oja** | 0.00468 ± 0.00025 | 0.838 ± 0.06371 | 0.312 ± 0.00112 | 19 |
| **LSTM** | 0.05190 ± 0.02175 | <u>0.778 ± 0.08576</u> | <u>0.151 ± 0.01259</u> | 13 |
| **GRU** | 0.03327 ± 0.01684 | **0.721 ± 0.01895** | **0.146 ± 0.00788** | <u>10</u> |
| **mean-HAG** | <u>0.00320 ± 0.00046</u> | 0.783 ± 0.00379 | 0.285 ± 0.00377 | **8** |
| **variance-HAG** | 0.01627 ± 0.01416 | 0.784 ± 0.00262 | 0.286 ± 0.00343 | 15 |

Bold marks the best (lowest) error; underlined values are the second-best.

**Table 3 | Dataset details showing sequence lengths, dimensions, and number of classes**

| Dataset | Avg. Size | Pretrain Size | Train Size | Test Size | # Dim x # Duplicates | # Classes |
|---|---|---|---|---|---|---|
| Japanese Vowels | 15.56 | 4,274 | 4,274 | 5,687 | 12 x 42 | 9 |
| CatsDogs | 148.0 | 24,272 | 24,272 | 24,272 | 20 x 25 | 2 |
| FSDD | 35.50 | 18,018 | 72,763 | 33,729 | 20 x 25 | 10 |
| Spoken Arabic Digits | 39.81 | 20,082 | 263,224 | 87,029 | 13 x 39 | 10 |
| Speech Commands | 157.91 | 78,835 | 14,969,965 | 1,741,325 | 20 x 25 | 35 |

HAG is a competitive approach in classification domains, where increased dimensionality and reduced feature redundancy directly benefit linear separability. GRU and LSTM require larger datasets to avoid overfitting, whereas RC models deliver better accuracy on smaller datasets. HAG is able to provide competitive or intermediate accuracy for the bigger datasets with a fraction of the computational effort. Unlike classification tasks where feature decorrelation aids linear separability, prediction tasks require a balance between preserving temporal dependencies and expanding the feature space: while adaptive connectivity can amplify the expressive power of reservoir states, it may also disrupt the stable temporal representations crucial for long-range forecasting. This may explain why the benefits of HAG are less pronounced for time-series forecasting.

Since our algorithms perform only marginally better compared to other models for forecasting tasks, we will focus our analysis on classification tasks, that constitute the main core of our theory and improvement over other methods.

In the next parts, we combine different approaches to gain a better picture of the reservoir's representational capacity. A central premise of reservoir computing is that mapping inputs into a sufficiently high-dimensional space increases the likelihood of linear separability (Cover's Theorem). In practice, however, there are two distinct notions of "dimensionality":

- Task-agnostic expressivity. How spread out the states are overall, irrespective of labels. This is what we probe first.
- Task-relevant effective dimensionality. How many directions actually help separate classes, i.e., large between-class variation with small within-class variation. This is what our separability metrics target.

Finally, no single metric fully characterizes the "richness" of reservoir states[37]. A single scalar value (e.g., spectral radius or average correlation) cannot capture all the ways a reservoir might fail or succeed at generating a rich, decorrelated state space. Multiple metrics, taken together, more reliably show whether the reservoir supports the nonlinear expansion and separability that underpins successful learning.

## Dimensionality of reservoir states

A tempting idea to estimate "directions" the reservoir states actually occupy is to calculate the matrix rank of reservoir states and treat that as the dimensionality measure. However, the rank merely tells us the maximum number of linearly independent vectors, it does not reveal how the variance or information is distributed across those dimensions. For instance, a reservoir whose state matrix has full rank might still concentrate most of its variance in just a few principal components (i.e., most singular values remain tiny). By definition, the rank also fails to distinguish between a matrix that robustly spans a high-dimensional space and one that merely has many near-redundant dimensions, ergo small amounts of noise can artificially inflate the rank.

Consequently, we adopt four complementary metrics: the *spectral radius* of the reservoir's weight matrix, the *average pairwise correlation* between neuron activations, and the *cumulative explained variance dimension* (CEVD) derived from principal component analysis (PCA) and *distance correlation*, a dependency measure between

random vectors. Each metric offers a different perspective on whether the reservoir supports sufficiently diverse and decorrelated dynamics.

Detailed values for every dataset/function combination are presented in Fig. 2.

**Spectral radius.** The spectral radius of the reservoir connectivity matrix is a fundamental parameter influencing the memory capacity and stability of ESNs. A higher spectral radius can allow for richer dynamics but must be carefully controlled to maintain the echo state property[38].

Across all classification tasks, variance-HAG systematically boosts the reservoir's spectral radius compared to other ESNs. Mean-HAG remains higher than static reservoirs but generally in the same range than local rule ESNs. For example, on Japanese Vowels the excitatory ESN sits at 0.72, while mean-HAG and variance-HAG raise it to 0.98 and 1.24, respectively. Similar gains appear on CatsDogs (0.82 →1.01/1.67), FSDD (0.97 →1.01/2.89), and Speech Commands (1.00 →0.98/1.99). By stretching the largest eigenvalue, HAG enlarges the reservoir's dynamic range and "memory horizon," creating richer state trajectories without destabilizing the network.

**Linear correlation.** To elucidate the dynamics of our reservoir, we assess the correlation among neural states[39]. For each experiment, we computed the linear coefficient between every pair of neuron time-series, $r_{ij} = corr(x_i, x_j)$, and report the mean of the absolute values $|r_{ij}|$. This statistic quantifies *redundancy*: a value of 1 signals perfect synchrony (either in phase or in antiphase), while 0 indicates fully independent trajectories.

Excitatory-only networks (E-ESN and both HAG variants) naturally generate very little anti-phase activity, so their $|r|$ is dominated by positive correlations. Signed reservoirs, on the other hand, produce positive and negative pairs in roughly equal numbers; if one averaged the raw $r_{ij}$ they would cancel and yield a misleading value near zero. Thus taking absolute values provides a more useful comparison across the two connectivity regimes.

Static excitatory reservoirs exhibit high redundancy–average correlations exceed 0.4 on every dataset and even approach 1.0 on complex tasks like Speech Commands. While being also based on excitatory only connection, mean-HAG and variance-HAG slash these figures dramatically: Japanese Vowels drop from 0.445 to 0.081/0.065; FSDD from 0.890 to 0.329/0.169; and Speech Commands from 0.994 to 0.570/0.476. A fully signed ESN (positive and negative weights) attains the lowest raw mean correlation because positive and negative synchrony partly cancel each other. However, when we consider $|r|$– which penalizes both highly positive and highly negative lock-step activity–HAG closes most of this gap without relying on inhibitory weights. In other words, long-horizon Hebbian rewiring is almost as effective as introducing explicit inhibition for the purpose of decorrelating the code.

**Cumulative explained variance dimensionality.** Next, we analyze the reservoir feature space expansion. Intuitively, if the reservoir can represent data in a higher-dimensional space, then downstream linear classifiers or predictors should have an easier time separating different classes.

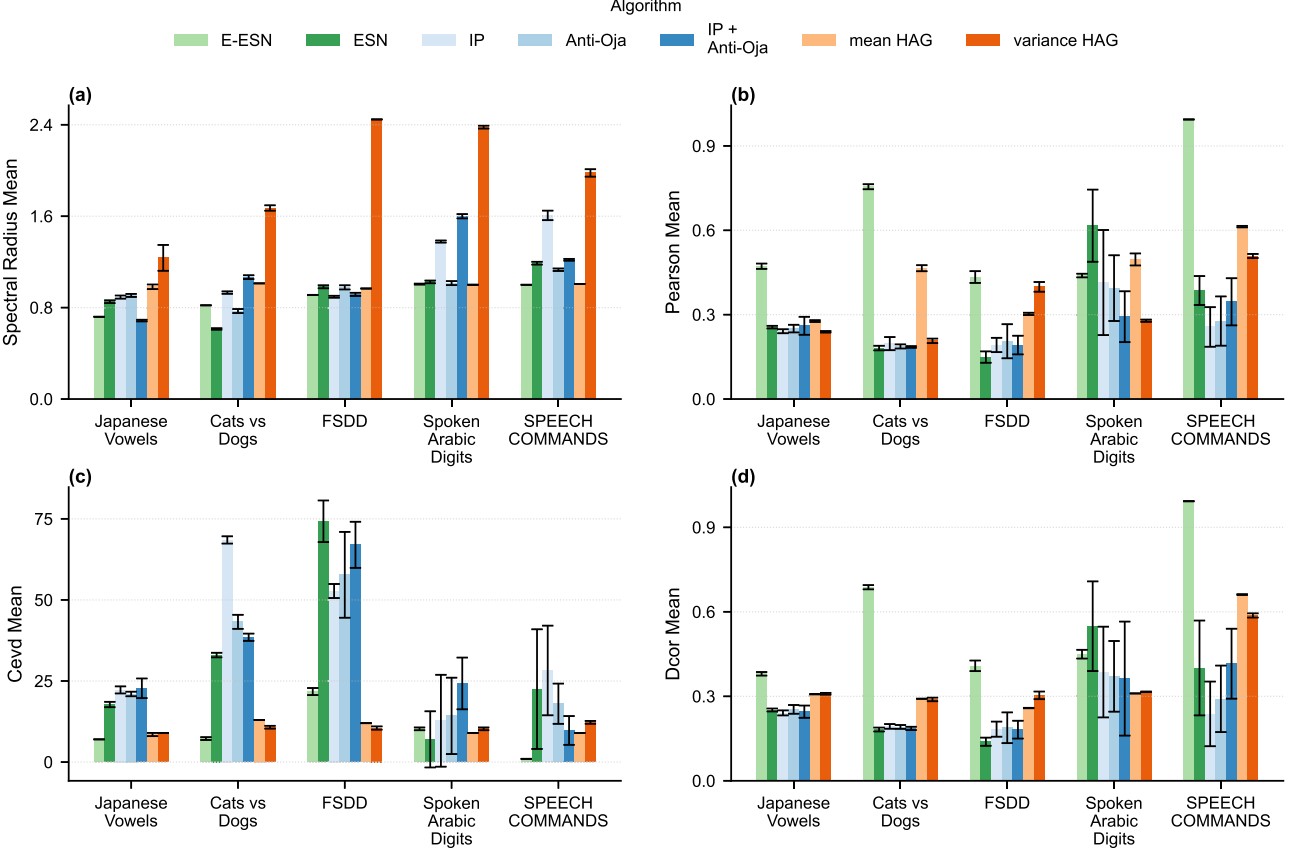

**Fig. 2 | Dimensionality metrics across all dataset-function combinations.** Each bar panel summarizes a complementary indicator of dynamical richness: (**a**) spectral radius of the recurrent weight matrix; (**b**) mean absolute pairwise correlation between neuron activations; (**c**) effective dimensionality via cumulative explained variance (CEVD); (**d**) distance correlation capturing nonlinear dependencies. Error bars represent standard deviation.

As an indication of the volume spanned by the reservoir states, we perform PCA on **H** (or equivalently on **H**$^\top$) to obtain singular values $\sigma_1 \geq \sigma_2 \geq \cdots \geq \sigma_n$. Each $\sigma_j^2$ is proportional to the variance captured by the $j$-th principal component. The cumulative explained variance up to the $d$-th principal component is then given by:

$$C_d = \frac{\sum_{j=1}^{d} \sigma_j^2}{\sum_{k=1}^{n} \sigma_k^2} \quad (3)$$

This cumulative measure indicates the total proportion of variance captured by the first $d$ principal components. To assess the effective dimensionality of the reservoir's state space, we determine the minimum number of principal components required to reach a predetermined threshold $\theta = 0.9$ of cumulative explained variance:

$$D = \arg\min_d (C_d \geq \theta) \quad (4)$$

A higher value of $D$ suggests that more principal components are needed to capture the same amount of variance, and can reflect more varied or expansive dynamics in the reservoir.

Results show again that HAG generally expands the reservoir's "effective" subspace compared to excitatory only ESNs. On CatsDogs, this rises from 7.25 components in the excitatory ESN to 13.0 (mean-HAG) and 10.75 (variance-HAG). For FSDD, it jumps even more–from 1.5 to 12.0/14.75–while on Speech Commands it climbs from 1.0 to 9.0/13.0. The only exception is Spoken Arabic Digits, where mean-HAG slightly lowers CEVD (10.25 → 9.0) and variance-HAG holds it steady (10.25). Overall, HAG broadens the reservoir's feature space–particularly for high-dimensional inputs–facilitating easier

linear separation but it again remains under the space of excitatory-inhibitory connections ESNs.

**Distance correlation.** In addition to linear measures of redundancy, we estimate *distance correlation*[40] among neuron activations to capture nonlinear dependencies using the official implementation of ref. [41]. Distance correlation is zero if and only if two random vectors are statistically independent, and thus serves as a strict test for any form of coupling. Given two sets of reservoir-state samples, $X = \{x_i\}_{i=1}^{n}$ and $Y = \{y_i\}_{i=1}^{n}$, we first form their pairwise Euclidean-distance matrices $A_{ij} = \|x_i - x_j\|$ and $B_{ij} = \|y_i - y_j\|$. After double-centering each ($\tilde{A}_{ij} = A_{ij} - \bar{A}_{i.} - \bar{A}_{.j} + \bar{A}_{..}$, likewise for $\tilde{B}$), the empirical distance covariance is $\mathrm{dCov}(X, Y) = \frac{1}{n^2} \sum_{i,j=1}^{n} \tilde{A}_{ij} \tilde{B}_{ij}$, and the distance correlation is $\mathrm{dCor}(X, Y) = \frac{\mathrm{dCov}(X, Y)}{\sqrt{\mathrm{dCov}(X, X)\, \mathrm{dCov}(Y, Y)}}$. In our analysis, we compute distance correlation for every pair of neuron-activation time-series and report their average. Low distance correlation indicates that neurons explore truly independent dimensions of state space–even nonlinearly–whereas high values reveal hidden synchrony or redundancy that linear correlation would miss.

Static excitatory ESNs showed a high average distance correlation, indicating persistent hidden couplings. It's normal that networks exhibiting high linear correlation also show high distance correlation. However, interestingly, when taking into account all forms of statistical coupling HAG matches or beats excitatory-inhibitory networks. This dramatic drop confirms that HAG promotes independent neuron trajectories–linear and nonlinear–despite using only excitatory connections.

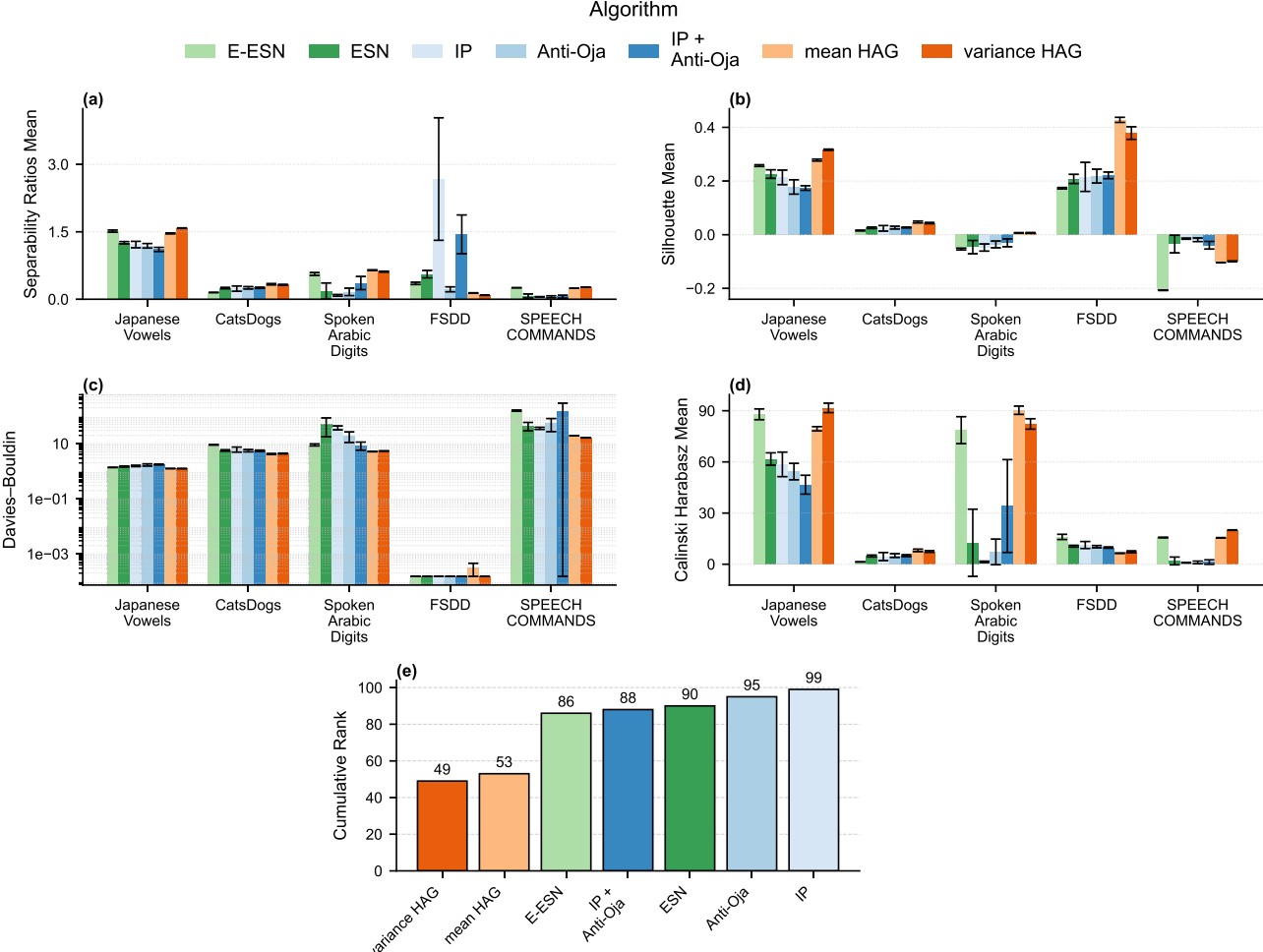

**Fig. 3 | Cluster-quality metrics for final reservoir states.** For each dataset and reservoir design we report: (**a**) inter/intra-class distance ratio, (**b**) silhouette score, (**c**) Davies-Bouldin index, (**d**) Calinski-Harabasz index, and (**e**) the cumulative rank across all 20 dataset-metric pairs. Higher values in (**a**, **b**, **d**) and lower in (**c**, **e**) indicate better separability and internal consistency. Error bars represent standard deviation.

## Separability and consistency of reservoir representations

A reservoir's ability to support accurate classification hinges not only on its raw dimensionality, but on how distinctly its states corresponding to different classes cluster in that space. In particular, we require (1) low variability within each class–so that points from the same label form tight, coherent clusters–and (2) high variability between classes–so that different labels occupy well-separated regions. Together, these properties ensure that a simple linear readout can draw boundaries that reliably discriminate among classes. To quantify this behavior, we evaluate multiple complementary metrics that capture both intra-class cohesion and inter-class separation of the final hidden-state representations.

1.  Inter-class vs. Intra-class distance ratio. We compute the average pairwise distance between class centroids (inter-class distance) and divide it by the average pairwise distance within each class (intra-class distance). The resulting ratio is given by: $\frac{\text{average inter} - \text{class distance}}{\text{average intra} - \text{class distance}}$ A higher ratio indicates centroids are far apart relative to each cluster's internal spread, suggesting tighter clusters and clearer separation.

2.  Silhouette score. For each point, the silhouette score measures how much closer it is to points within its own cluster compared to points in the nearest other cluster. Formally, the silhouette score for each point i is defined as:

$$s_i = \frac{b_i - a_i}{\max(a_i, b_i)}, \qquad (5)$$

where $a_i$ is the mean intra-cluster distance and $b_i$ is the mean distance to the nearest alternative cluster. Averaging $s_i$ over all points gives a value in the range [-1,1], with values closer to 1 indicating better separation[42].

3.  Davies-Bouldin index (DBI). This index is calculated as the average ratio of within-cluster scatter to between-cluster separation across all clusters. Lower DBI values correspond to clusters that are compact and well separated[43].

4.  Calinski-Harabasz index. Also known as the variance ratio criterion, this measure is the ratio between the between-cluster dispersion and the within-cluster dispersion, scaled by the number of clusters and data points. Higher values indicate well-separated, compact clusters[44]. By reporting multiple clustering metrics, we obtain a nuanced view of how different reservoir architectures balance the compactness of within-class representations and the separation between different classes. Figure 3 summarizes these results across all dataset-reservoir combinations.

Across the five speech-and-audio benchmarks, either mean-HAG or variance-HAG attains the best scores in 13 of the 20 dataset × metric combinations, and at least one of the HAG algorithm performs best on 17 of the 20 dataset × metric combinations. Consistently, the HAG variants also achieve the best (lowest) cumulative rank across all models.

E-ESN is a strong baseline on those metrics, being consistently the closest challenger. On Japanese Vowels and FSDD the raw excitatory-

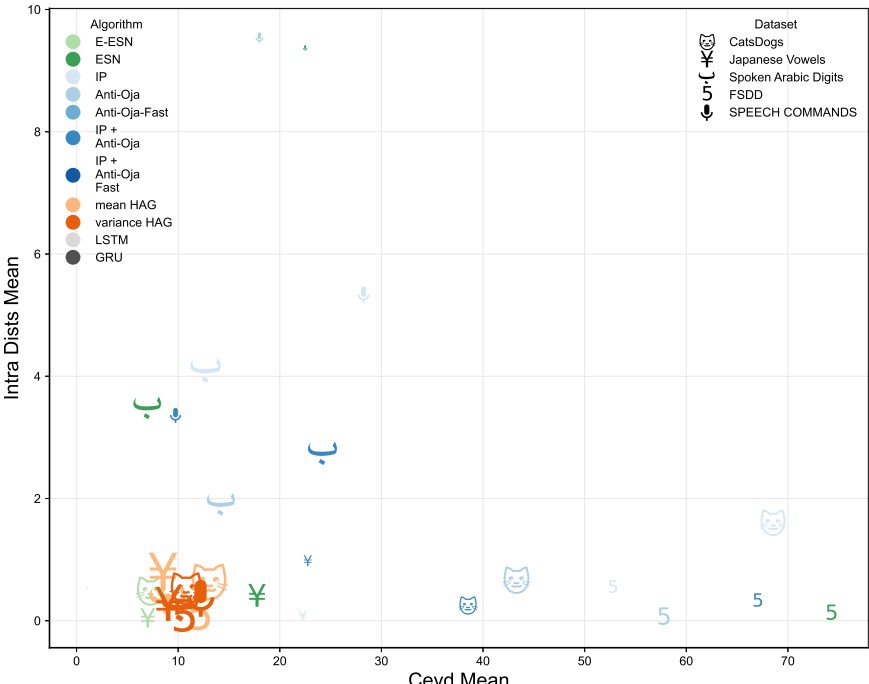

**Fig. 4 | Trade-off between reservoir dimensionality and separability across algorithms.** Mean CEVD (horizontal axis, *CEVD Mean*) against the *mean intra-class Euclidean distance* (vertical axis, *Intra Dists Mean*); marker size encodes the final test accuracy and color/symbol identify the algorithm/dataset combination.

only network already forms tight clusters and mean-HAG can merely match it, while variance-HAG edges ahead by a small margin. The other exception is the large-scale Speech Commands corpus, where all reservoirs still struggle to achieve clean separation: variance-HAG performs better on the Davies-Bouldin index, raises the Calinski-Harabasz but the silhouette score remains better for classic ESNs and local plasticity rules ESNs.

Networks trained only with Intrinsic Plasticity or Anti-Oja rules never surpass HAG and seldom exceed even the static ESN; short-term weight updates do not reorganize the latent space as effectively as the long-horizon Hebbian growth employed by HAG.

In summary, both HAG variants produce reservoirs whose hidden states are, for almost every dataset and metric, both more compact within each class and more widely separated between classes. These results confirm that wiring neurons which co-activate over long time windows is an effective strategy for carving out linearly-separable manifolds in high-dimensional, real-world data.

Taken together, these results confirm that HAG, by dynamically connecting co-active neurons, produces reservoirs with significantly improved linear separability compared to static excitatory and standard plasticity-based reservoirs. Both mean- and variance-HAG consistently deliver superior intra-class compactness and inter-class separation, underpinning their demonstrated performance advantages in classification tasks.

### Synthesis: dimensionality-separability trade-offs

Overall, these results confirm that HAG not only expands the reservoir's effective dimensionality compared to excitatory network (cf. Section Dimensionality of reservoir states), but also organizes the state space into well-defined, linearly separable clusters—thereby underpinning its superior classification accuracy (cf. Section Results).

Both variants of HAG consistently exhibit spectral radii and effective dimensionality similar or lower to fully signed reservoir reservoirs, including reservoir tuned with local plasticity rules. At the same time, the Hebbian rewiring mechanism markedly lowers pairwise correlations among neuron activities compared with excitatory-

only baselines, promoting genuine statistical independence and reducing redundancy even without inhibitory synapses.

Regarding separability, HAG reservoirs exhibit markedly improved clustering metrics compared to baselines. Across multiple tasks-datasets combinations.

HAG tends to produce near-optimal within-class compactness with only moderate CEVD, which indicates it learns a task-relevant subspace. Architectures that push CEVD higher do not necessarily improve accuracy, suggesting that beyond a dataset-dependent threshold, extra components mostly add noise. Thus, HAG organizes variance along discriminative directions rather than chasing the number of principal components.

Figure 4 condenses part of those results into a single, two-dimensional diagnostic of reservoir "behavior". A good reservoir should simultaneously expand the input into a high-dimensional feature space (large CEVD) and keep samples from the same class close together (small intra-class distance), so the theoretical ideal region of the plot is the lower-right quadrant. However, we see that the best performing reservoir, grown by Hebbian Architecture Generation occupy a small, well-defined region of the design space where class consistency is near-optimal but internal expressivity remains low compared to other methods. Architectures with higher CEVD are not necessarily performing the best, failing to translate that extra expressivity into better accuracy. This suggests that beyond a dataset-dependent tipping point, extra dimensions mainly inject noise and erode class structure. This analysis therefore leads to refine our claims: HAG favors a minimal-sufficient expansion that aids class structure, rather than an indiscriminate increase in dimensionality.

## Discussion

We introduced the Hebbian Architecture Generation (HAG) method, an adaptive approach to reservoir computing that dynamically constructs connectivity patterns based on long term neuron correlation. HAG promotes decorrelation in neural states and separation between classes, shaping the network structure with the intrinsic statistical

properties of the input data, this results in HAG performing better than traditional static or plasticity based reservoir.

However, while HAG consistently outperforms baseline ESN methods in classification tasks by optimizing feature separability, its performance in prediction tasks is less decisive. The mechanisms that drive connectivity adaptation in HAG may be better suited to tasks requiring distinct feature separation rather than continuous temporal forecasting.

Despite this, HAG provides valuable insights into adaptive reservoir design. By dynamically structuring the network to fit the task, it bridges the gap between static ESNs and fully trainable recurrent models. Importantly, HAG achieves these results through a single, unsupervised pass without introducing inhibitory synapses.

In summary, this approach offers a practical and biologically motivated strategy for developing high-performance reservoirs, preserving the simplicity of linear readouts and narrowing the observed performance gap with fully trained recurrent neural networks on the evaluated benchmarks.

Future research should further explore the scalability of HAG, its effectiveness in larger and more complex tasks, and possible modifications to enhance its predictive modeling capabilities. By refining biologically inspired plasticity mechanisms, HAG paves the way for more robust and adaptable neural computing architectures.

## Methods

Below, we detail the two algorithms central to this work. Pseudo-code for each is provided in Appendix A.1. Throughout this paper, our inputs are multivariate time-series: each time-step $\mathbf{u}[t] \in \mathbb{R}^d$ carries $d > 1$ simultaneously recorded signals. All of our algorithmic developments and evaluations therefore assume and exploit this multidimensional structure. As part of preprocessing, all input signals are normalized to the [0,1] range, ensuring that values remain non-negative (consistent with the excitatory-only nature of synaptic weights used in our model).

### HAG algorithm

The network is initialized as a blank reservoir containing no recurrent connections, except for fixed input links. Input weights are drawn from a uniform distribution, $W_{in} \sim \mathcal{U}(0,1)$, and neuron biases from a normal distribution, $b_i \sim \mathcal{N}(0.1, 0.1)$. HAG dynamically modifies the synaptic weights $w_{ij}$ whenever a neuron's measured activity (i.e., its mean or variance) deviates from a predefined homeostatic range.

By design, the reservoir is restricted to positive (excitatory) synapses ($w_{ij} > 0$). This choice aligns with our biological inspiration of excitatory synaptogenesis and simplifies the weight update rule. However, limiting connections to excitation can reduce the reservoir's computational richness.

While limiting connections to excitation could reduce computational richness, it markedly simplifies implementation on substrates that naturally realize non-negative weights[45–47]. The reason is that when signed interactions are required, such systems can still recreate effective negative weights via differential/balanced branches or two-device encodings that implement $w = w^+ - w^-$ doubling the size of the implementation. We discuss this perspective further in Appendix A.3. In this paper, we deliberately keep the recurrent graph strictly excitatory and retain only a conventional signed linear readout.

### Homeostatic control of plasticity.

In neural systems that rely on Hebbian-like "fire together, wire together" rules, a common failure mode is runaway growth: if neurons frequently co-activate, their connections can strengthen uncontrollably until they saturate. Homeostatic plasticities are mechanisms that act to stabilize the activity of a neuron around some set-point value. Several forms exist[48,49], and some Hebbian rules, such as Oja's rule and the BCM rule, incorporate a homeostatic component directly into their update dynamics. However,

our method is inspired by Homeostatic Structural Plasticity (HSP)[29,50–52], which operates at the level of individual synaptic strengths.

We propose two variants, each linked to a different homeostatic mechanism:

1.  mean-HAG: corrects deviations from a target *mean* firing rate.
2.  variance-HAG: corrects deviations from a target *standard deviation*. While mean-based regulation enforces an overall baseline firing rate, variance-based regulation ensures that neurons retain a wider dynamic range. Since we do not have a formal proof favoring one approach universally, these two variants allow us to compare the effects of emphasizing average activation versus activation variability, and to investigate which is more beneficial for a given task.

### Identifying neurons that are not at homeostasis.

We write $\kappa \in \{r, v\}$ to distinguish the rate- and variance-controlled versions. Every $T_{current}$ time steps we compute, for each neuron $i$,

$$\Delta z_i = \frac{1}{\beta_\kappa}(s_i - \rho_\kappa) \tag{6}$$

1.  mean-HAG ($\kappa = r$): $s_i = \langle x_i \rangle_T$ is the average firing rate of neuron $i$ over the last $T$ steps. $\rho_r$ is the target mean rate, and $\beta_r$ sets the permissible deviation ("rate spread").
2.  variance-HAG ($\kappa = v$): $s_i = \sigma_{x_i, T}$ is the sample standard deviation of neuron $i$'s activity over the same window. $\rho_v$ is the target standard deviation, and $\beta_v$ is the corresponding spread. A neuron is considered under-active if $\Delta z_i < -1$ and over-active if $\Delta z_i > +1$; synapses are grown or pruned accordingly.

If $\Delta z_i < -1$, the neuron needs to increase its activity. In this case, one incoming connection weight is increased by $\delta w$. The creation of new connections is restricted to neurons that have been identified as requiring additional connections. To choose which connection to increase, for every neuron that has not yet achieved homeostasis, we compute pairwise linear correlation coefficients[39] with every other neuron that is also not at homeostasis and we establish an incoming connection with the highest correlated neuron. A comparable homeostatic mechanism applied to randomly generated connections has already been shown to induce Hebbian-like structure[29,51], reinforcing the biological plausibility of our approach.

If $\Delta z_i > 1$, the neuron needs to decrease its activity. In this case, one connection weight is decreased by $\delta w$. Unlike the creation of new connections, the pruning of connections is performed randomly, independently of the state of the neuron's partners and regardless of whether they also need to decrease their activity. This is because we lack a reliable local criterion for removal that increases global dimensionality. Specifically, we draw $j$ uniformly from the current connected neurons $\{j | w_{ij} > 0\}$ and update $w_{ij} \leftarrow \max(0, w_{ij} - \delta w)$

The window size $T_{current}$ itself is randomly sampled at each adaptation step from a logarithmically spaced grid between $T_{min}$ and $T_{max}$ which are hyperparameters. By sampling randomly from these intervals, the algorithm robustly captures correlations occurring across multiple time scales, rather than relying exclusively on a single fixed temporal resolution. A FULLINSTANCE mode also exist in which $T_{current}$ is set to the length of the next instance.

### Identifying the most correlated pair.

To form new connections, we consider only neurons $i$ for which the growth indicator $\Delta z_i$ satisfies :

$$\Delta z_i < -1. \tag{7}$$

For this subset of neurons, we compute pairwise correlation coefficients $r_{ij}$ for all pairs, ($r_{ij}$ definition is recalled in Appendix A.2).

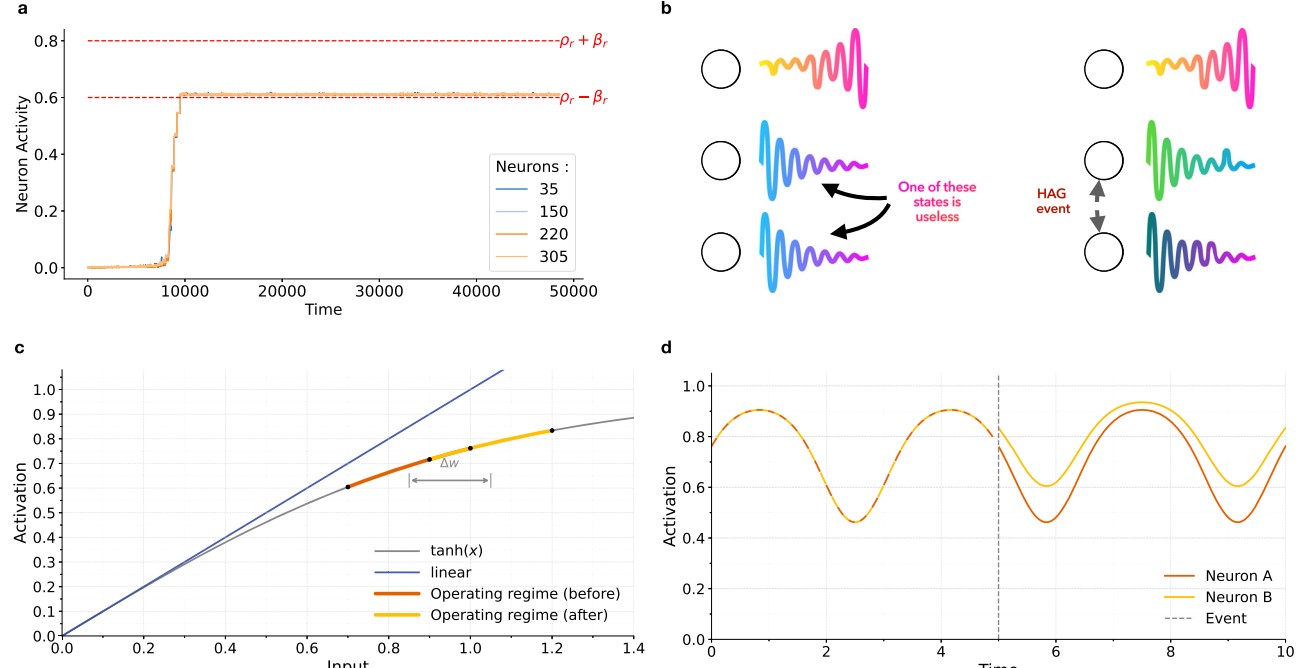

**Fig. 5 | HAG reshapes reservoir connectivity and dynamics. a** Different neurons' activities during training with the mean-HAG algorithm. **b** Initially correlated nodes create a connection. **c** Input-output mapping showing how an additive $\delta w$ shifts operating points to different regions of the nonlinearity. **d** Time-series trajectories of two neurons diverging at the moment of the HAG event (vertical dashed line). In all panels, the gray arrow or dashed line marks the HAG event itself.

The pair $(i^*, j^*)$ with the highest correlation is selected:

$$(i^*, j^*) = \arg\max_{(i,j)}(r_{ij}), \tag{8}$$

where the maximization is therefore performed over all neuron pairs $(i, j)$ that have not yet achieved homeostasis.

**Establishing the connection.** Once the most highly correlated pair $(i^*, j^*)$ is identified, we strengthen the incoming synapse to the under-active neuron $w_{i^*j^*} \leftarrow w_{i^*j^*} + \delta w$, where $\delta w > 0$ is the increment step.

This mechanism ensures that connections are formed preferentially between neurons that exhibit high correlation, promoting the restructuring of the reservoir to enhance its dynamic representation of input data.

Although updates are applied sequentially, in practice the resulting edge is reciprocal. The reason is that $r_{ij} = r_{ji}$ and, when $j^*$ is later processed as a target, it faces the same candidate set and will select $i$ unless three neurons have the same correlation which is unlikely. Hence, except for rare tie-breaks, $(i, j)$ becomes bidirectional within the same sweep. Asymmetry arises only when random pruning removes one direction but not the other.

**Convergence.** The network is said to be at homeostasis if, for each neuron $i$, $\Delta z_i$ is between -1 and 1 (i.e., $s_i$ is between $\rho_\kappa - \beta_\kappa$ and $\rho_\kappa + \beta_\kappa$). At homeostasis the network maintains a desired level of variance or average in neuronal activity as seen in Fig. 5.

In mean-HAG, maintaining the target mean activity directly promotes the network stabilizes as seen in Fig. 5. In variance-HAG, which increases variability (and potentially signal strength), we add a homeostatic safeguard: if any neuron's state exceeds a saturation threshold $\theta_{\text{sat}}$, we scale its synaptic weights down by a factor $\eta_{\text{sat}}$. This synaptic scaling[53] mechanism keeps the network in a balanced regime, promotes stability in practice and prevents blow-up.

Formal convergence is not guaranteed and due to tie-breaks and random pruning, the final configuration is not unique and may vary across runs on the same series.

### Datasets
ESNs have employed a diversity of benchmarks and datasets, as extensively documented in ref. 54. To test our algorithm, we select a representative suite drawn from that spectrum.

We utilized `ReservoirPy`[55], a library updated with contemporary advancements, featuring a modular architecture for assembling ESNs and a suite of standard algorithms for training the readout layer.

### Task types
The training of an ESN system follows a two-step process: first, the reservoir processes input signals into high-dimensional state representations, and second, a readout layer is trained to map these representations to the desired outputs.

For classification tasks (e.g., speech recognition), the reservoir processes the entire input sequence and retains its final state as a compact summary of the sequence dynamics. This final state serves as the feature vector for training the readout layer $W_{out}$ to correctly classify the input.

For prediction tasks, where the goal is to forecast future values (5 steps ahead) from past inputs, we adopt a sequence-to-sequence approach. Unlike classification, where only the final state is used, here the readout layer $W_{out}$ is optimized to minimize the Normalized Root Mean Square Error (NRMSE) between the predicted outputs and the actual targets at each time step.

### Classification datasets
**Japanese vowels.** The Japanese Vowels dataset includes recordings of nine male speakers pronouncing sequences of Japanese vowels. It is frequently utilized in research on linguistic characteristics and speaker identification technologies[56].

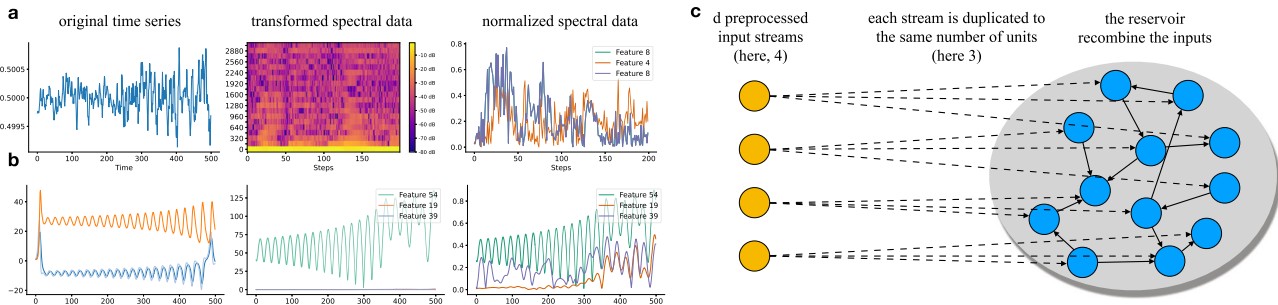

**Fig. 6 | Experimental setup and preprocessing pipelines. a** For classification tasks, raw audio signals are converted to Mel-Frequency Cepstral Coefficients (MFCCs); **b** for forecasting tasks, signals undergo Short-Time Fourier Transform (STFT) to preserve temporal structure; **c** each reservoir neuron receives one input channel to ensure balanced coverage of the input space.

**CatsDogs.** The CatsDogs dataset is the auditory counterpart to the classic image classification task, containing WAV audio files - 164 for cats (1,323 seconds) and 113 for dogs (598 seconds) - recorded at 16 kHz[57].

**FSDD.** The Free Spoken Digit Dataset (FSDD) is an open collection of English audio recordings of spoken digits 0–9 by multiple speakers. Designed for experimenting with speech processing techniques like classification and clustering, it provides a straightforward entry point into digital speech processing[58].

**Spoken arabic digits.** The Spoken Arabic Digits dataset contains recordings of 88 individuals pronouncing Arabic digits 0-9, with ten pronunciations per digit per speaker. It is commonly used for testing speech recognition algorithms due to the phonetic diversity of Arabic numerals[59].

**Speech commands.** The Speech Commands dataset comprises over 105,000 audio files of short commands like "Yes," "No," "Up," and "Down," spoken by various speakers. Widely used for training and benchmarking models in voice user interfaces[60].

### Prediction datasets
**Mackey-Glass.** Derived from a differential equation, the Mackey-Glass dataset is noted for its use in modeling nonlinear dynamics and chaos, making it a challenging dataset for time-series prediction models[61].

**Lorenz.** The Lorenz dataset is based on the Lorenz attractor, a set of chaotic differential equations used extensively in predicting nonlinear system behaviors and atmospheric studies[62].

**Sunspot daily (SILSO).** The Sunspot dataset from SILSO includes smoothed daily sunspot numbers from 1749 to 2020, reflecting solar activity and serving as a proxy for the Sun's magnetic field strength. Its complexity makes it a significant test case for forecasting models in time-series analysis and solar studies[63].

### Preprocessing
Because our goal is multivariate forecasting and classification, we preprocess each input channel appropriately (MFCC, STFT, etc.) so that the reservoir sees a $d$-dimensional vector at each time step. These transformations serve to highlight the most informative temporal and frequency-domain patterns, making it easier for the reservoir to process and learn from the data.

We employ different preprocessing techniques tailored to classification and prediction tasks, each designed to transform raw time-series data into meaningful representations for model training.

### Preprocessing for classification tasks
For classification tasks, we convert audio signals into spectral representations using *Mel-Frequency Cepstral Coefficients* (MFCCs), computed with the `librosa` library[64]. MFCCs are widely used in speech and audio processing as they capture the spectral characteristics of a sound while approximating the human auditory system's response. It is standard practice in reservoir computing classification[7,65,66]. The classification preprocessing steps are illustrated in Fig. 6.

Note that Japanese Vowels and Spoken Arabic Digits dataset are already provided in MFCC format, so we reuse the given features directly.

### Preprocessing for forecasting tasks
For forecasting tasks, we convert signals into spectral representations using *Short-Time Fourier Transform* (STFT). This method isolates relevant frequency components while preserving temporal structure, ensuring that key predictive patterns remain intact. Unlike classification preprocessing, we maintain the original number of time steps, which is crucial for sequence-to-sequence forecasting. The prediction preprocessing steps are shown in Fig. 6.

### Input normalization
Before feeding the features into the reservoir we scale every input dimension to [0, 1] with a Min-Max transform, $\mathbf{u}' = \frac{\mathbf{u}-\min(\mathbf{u})}{\max(\mathbf{u})-\min(\mathbf{u})}$, where the minima and maxima are estimated only on the training split and re-applied unchanged to validation and test data.

### Size standardization
To ensure comparability across heterogeneous datasets, we standardize reservoir size using a duplication rule. As shown in Fig. 6, we take our d preprocessed input streams and duplicate each one k times:

$$ k = \left\lceil \frac{500}{d} \right\rceil, \qquad n = k\,d, $$

so that all models (standard ESNs, excitatory-only ESNs, IP- or Anti-Oja-adapted reservoirs, and HAG variants) use approximately $n \approx 500$ neurons (e.g., Spoken Arabic Digits: 13 × 39 = 507; Japanese Vowels: 12 × 42 = 504; see Table 3).

This guarantees that each reservoir neuron receives exactly one input channel, enforcing an even "fair share" of the input space before the recurrent mixing that produces the high-dimensional reservoir state. The sensitivity to this technique is described in Appendix D.

### Compared models and training procedures
We evaluate the following five model families:
(a)  Excitatory-only ESNs (E-ESN): reservoirs with only positive recurrent weights,

(b) Signed ESNs: standard ESNs with both positive and negative recurrent weights,

(c) Local plasticity ESNs: IP as described in ref. 18; Anti-Oja with synaptic normalization and IP + Anti-Oja with synaptic normalization generated ESN as both refs. 22 and [23] showed that IP + Anti-Oja led to the best performances among the plasticity rules they evaluated,

(d) Gradient-trained : a gated recurrent unit (GRU)[67] and a long short-term memory (LSTM)[68] network, included to contextualize when lightweight reservoir adaptation is competitive versus conventional end-to-end gradient-trained recurrent network,

(e) Our adaptive HAG variants: mean-HAG and variance-HAG.

**Unsupervised pretraining.** We use a single unsupervised pretraining stream for the models that need it. The unsupervised pretraining sequence of length $L_{pre}$ is formed by concatenating all training examples (full training set for every dataset except Speech Commands). For Speech Commands we subsample 500 utterances (stratified by class) to cap computational cost while preserving acoustic diversity. During pretraining, only the adaptive reservoirs (HAG, IP, Anti-Oja, IP+Anti-Oja) update internal weights; static ESN / E-ESN and the LSTM baseline do not see an extra unlabeled pass beyond their supervised training folds.

**Readout training.** All readouts are trained with ridge regression (classification: sequence-to-vector, prediction: sequence-to-sequence) except for LSTM/GRU, which is optimized end-to-end with Adam.

**LSTM/GRU training.** The GRU and LSTM baseline are single-layer with hidden size $n$. This choice is conservative in favor of the LSTM and GRU, since a single-layer RNN with hidden size $n$ contains approximately $g(dn + n^2 + n)$ trainable parameters ($g = 4$ gates per cell for LSTM and $g = 3$ per cell for GRU), compared to $dn + pn^2 + n$ for a sparsely connected ESN (density $p$). Thus, even with moderate $p$, the LSTM contains significantly more trainable degrees of freedom. A parameter-matched LSTM would require a reduced hidden size $\tilde{n} \approx \sqrt{p/4}\, n$; for instance, if $p = 0.1$, then $\tilde{n} \approx 0.16\, n \approx 80$. By using the same hidden size $n$, we bias the comparison in favor of the gradient baseline, making HAG's performance more noteworthy. However, we do allow the size of the network to be reduced to avoid overfitting. Because PyTorch's built-in dropout is inactive when $num_layers = 1$, we apply an explicit $nn.Dropout(p)$ to the last hidden state (or to the concatenation of forward + backward states when bidirectional). Classification uses a cross-entropy loss; forecasting is trained with MSE. Gradients are clipped to a global $L_2$-norm of 1.0 and all weights follow PyTorch's default orthogonal/Xavier initialization.

**Hyperparameter search**
To ensure robust model evaluation, we employed cross-validation strategies appropriate for each task type. For classification tasks without predefined groups, we used Stratified 3-Fold cross-validation with shuffling to maintain class distribution across folds. When group-based classification was necessary, Stratified Group 3-Fold cross-validation was applied to preserve both class distribution and group integrity, preventing data leakage. For time-series prediction tasks, time-series Split cross-validation was utilized to respect temporal ordering and prevent future data leakage.

Hyperparameter tuning was performed using `optuna`[69], leveraging the Tree-structured Parzen Estimator (TPE)[70] sampler over 400 trials per dataset and algorithm variant. The TPE sampler efficiently explores the hyperparameter space by focusing on promising regions, making it suitable for our optimization tasks[71,72]. Further justification for the choice of hyperparameter optimization algorithm is given in Appendix B.2.

**Table 4 | Dominant asymptotic time complexity per full experimental cycle (adaptation + supervised training + labeled inference), supposing that the length of pretraining contain all the training instances ($L_{pre} = N_{seq}\, \bar{L}$)**

| Method | Forward pass | Adaptation | Supervised Training |
|---|---|---|---|
| Static ESN / E-ESN | $\sim N_{seq}\, \bar{L}\,(p\, n^2 + n\, d)$ | – | $+ \sim N_{seq}\, \bar{L}\, n^2 + n^3$ |
| IP | | $+ \sim 10\, N_{seq}\, \bar{L}\, n$ | |
| Anti-Oja / IP +Anti-Oja | | $+ \sim 5\, N_{seq}\, \bar{L}\, p\, n^2.$ | |
| HAG | | $+ \sim N_{seq}\, \bar{L}\, n^2$ | |
| LSTM | $\sim 4\, E\, N_{seq}\, \bar{L}(n^2 + nd)$ | – | $+ \sim 5\, E\, N_{seq}\, \bar{L}(n^2 + nd)$ |
| GRU | $\sim 3\, E\, N_{seq}\, \bar{L}(n^2 + nd)$ | – | $+ \sim 4\, E\, N_{seq}\, \bar{L}(n^2 + nd)$ |

The search space is partitioned into five categories:

(i) Shared ESN parameters (all reservoir variants): input scaling $s_{in}$, bias scaling $s_b$, ridge coefficient $\lambda$.

(ii) Static reservoir parameters (ESN / E-ESN): connection probability $p$ (signed or excitatory-only), spectral radius $\rho_s$.

(iii) Local plasticity parameters (only for adaptive baselines): IP target mean $\mu$, IP target variance $\sigma_{ip}$; IP learning rate $\eta_{ip}$; Anti-Oja learning rate $\eta_{oja}$; the combined IP+Anti-Oja model searches the union.

(iv) HAG structural growth parameters: incremental weight step $\delta w$, optional max in-degree $\gamma$, adaptation window bounds ($T_{min}, T_{max}$), and either mean-homeostasis ($\rho_r, \beta_r$) (mean-HAG) or variance-homeostasis plus saturation ($\rho_v, \beta_v, \theta_{sat}, \eta_{sat}$) (variance-HAG).

(v) LSTM/GRU parameters: hidden size $h$, we use one recurrent layer to keep the comparison to the 1-layer reservoir framework meaningful, dropout $p_{drop}$, bidirectionality flag $b$, learning rate $\eta$, batch size $B$, and number of epochs $E$.

Full parameter ranges and discrete grids appear in Appendix B.1; selected best values are tabulated in Appendix B.3; cross-validation scores are summarized in Appendix B.4.

**Computational complexity and efficiency**
We derive detailed operation counts in Appendix C.2; the principal asymptotic results are summarized in Table 4 and a realistic estimation of computation costs is given in Appendix C.3.

**Notation.** $n$: number of reservoir neurons (and LSTM/GRU hidden size); $d$: input dimensionality; $p$: recurrent connection density (fraction of nonzero entries in $W$, with $p = 1$ if dense); $L_{pre}$: length of the (concatenated) unsupervised pretraining stream (used only by adaptive reservoirs); $N_{seq}$: number of labeled sequences; $\bar{L}$: mean labeled sequence length; $E$: number of supervised training epochs for LSTM/GRU;

**Interpretation.** The computational hierarchy is straightforward:

(a) Static ESN / E-ESN incur only the baseline propagation $\sim pn^2 + nd$; they provide the lowest cost but no adaptation.

(b) IP adds an $\sim n$ per-step overhead (running gain/bias updates), asymptotically negligible for moderate $p$; total cost is effectively the static baseline.

(c) Anti-Oja / IP+Anti-Oja pay an additional $\sim pn^2$ every timestep for synaptic updates. Although the order matches the forward pass, the constant factor (extra multiplies/adds per weight) makes these strictly more expensive in practice.

(d) HAG replaces continuous per-timestep synapse updates with sparse, event-driven bursts. Its cumulative overhead depends on the number of neuron not at homeostasis $s_e$. Empirically $s_e \ll n$ after an initial transient; nevertheless, our complexity bounds adopt the worst-case $s_e = n$. Under this assumption, the cumulative overhead is comparable in order of magnitude to Anti-Oja. Thus HAG achieves structural rewiring with a cost profile competitive with–and often better than–local synaptic plasticity rules.

(e) LSTM/GRU (BPTT + Adam) are fundamentally more expensive: every labeled timestep incurs both forward and backward gate computations, multiplied by the number of epochs $E$. This leads to one to two orders of magnitude more floating-point operations under typical settings.

Despite this large gap in compute, HAG narrows the performance difference to a fully trained recurrent model (Sections Results), preserving the classical reservoir advantages: single-pass unsupervised adaptation, closed-form readout, and no gradient propagation through time. Thus, HAG offers a favorable accuracy-efficiency trade-off: substantially richer representations than static or purely local-plastic reservoirs, while remaining far cheaper than multi-epoch gradient-based sequence models.

**Restrictions.** All datasets are subject to their original licenses/terms of use. No clinical or proprietary third-party data were used.

**Reproducibility.** The datasets and the data generated in this study can be found or recreated by running the publicly available code as described in the Code availability statement.

## Data availability
The data used in this study are either publicly available benchmark datasets or synthetic series generated from standard equations. The real-world speech and audio datasets are available in public repositories as follows: the Free Spoken Digit Dataset (FSDD) at Zenodo (https://doi.org/10.5281/zenodo.1342401); Speech Commands (version 0.02) via the torchaudio SPEECHCOMMANDS repository (https://docs.pytorch.org/audio/main/generated/torchaudio.datasets.SPEECHCOMMANDS.html); Spoken Arabic Digits (10.24432/C52C9Q); Japanese Vowels (10.24432/C5NS47); and CatsDogs (https://timeseriesclassification.com/description.php?Dataset=CatsDogs).
The Sunspot daily (v4.0) series used in this work is available from SILSO at Zenodo (10.5281/zenodo.4654722). The Mackey-Glass and Lorenz time series analysed in this study are generated from the standard equations; no external data are required, and the synthetic series can be fully regenerated from the accompanying code. All scripts needed to download the public datasets, generate the synthetic time series, and reproduce the processed feature matrices and train/validation/test splits are provided in the associated Code Ocean capsule (10.24433/CO.3241639.v1).

## Code availability
All code is available under an MIT License on CodeOcean https://doi.org/10.24433/CO.3241639.v1.

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

## Acknowledgements

This project has received funding from the European Union's Horizon 2020 research and innovation program under the Marie Skłodowska-Curie grant agreement No 860949.

## Author contributions

T.C. conceived the study, developed the methodology and algorithms, implemented the experiments, analysed the data and wrote the initial manuscript draft. J.D. provided supervision, guidance, and critical revision of the manuscript. All authors have read and approved the final manuscript.

## Competing interests

The authors declare no competing interests.
