## [Transparent Peer Review file · Nature Communications]

Reshaping reservoirs with unsupervised Hebbian adaptation

Corresponding Author: Mr Tanguy Cazalets

Version 0:

Reviewer comments:

Reviewer #1

(Remarks to the Author)

In this paper, the authors introduce Hebbian Architecture Generation (HAG), an unsupervised method for dynamically building task-specific reservoir networks by growing synaptic connections between neurons that frequently co-activate. Starting from a nearly empty Echo State Network, HAG incrementally reshapes connectivity using long-term correlations rather than immediate updates, inspired by biologically grounded principles. With two variants—mean-HAG and variance-HAG—this approach improves both classification and prediction performance across various real-world datasets, outperforming existing plasticity-enhanced reservoirs without relying on gradient-based learning. I think this work is valuable. After addressing my comments below, I recommend for publication.

COMMENTS

- Pag 6. “To choose which connection to increase, for every neuron that has not yet achieved homeostasis, we compute pairwise linear correlation coefficients (Pearson, 1895) with every other neuron that is also not at homeostasis and we establish an incoming connection with the highest correlated neuron.” Provide an intuitive explanation for this methodological choice. Why do you think that increasing by Δw this particular connection yields the most benefits?
- When decreasing a connection by Δw , the connection to prune is selected randomly. What is the rationale behind this choice?
- Is the HAG process a converging method? What can we say about the existence and uniqueness of synaptic configurations that satisfy the HAG requirements? Does the HAG process always converge to the same configuration for a given input time series?
- Eq. 5. Due to symmetry, if the Pearson coefficient is maximised for (i^*, j^*) then it is maximised for (j^*, i^*) . Then why one should increase by Δw only the connection (i^*, j^*) and not the connection (j^*, i^*) ? Are those connections increased both? If so the Figure 2 (b) is misleading.
- How to select the desired homeostatic values of ρ_k and β_k ?
- Fig 5 describes a modular organisation of the input injected in the reservoir. How sensitive are the performance on this particular modularisation technique? Sometimes ESNs struggle to process high-dimensional inputs due to interference dynamics. Is the same modular injection applied for standard ESN?
- The HAG method is claimed to be better than anti-Oja and Intrinsic Plasticity (IP), and this statement is supported by experiments using different metrics. However, the paper would benefit from a computational complexity analysis. Is the HAG computational time higher/lower than alternatives like IP or anti-Oja?
- A Table 1. Maybe better Train Size and Test Size instead of Train Length and Test Length.

• In pag 19 is reported "Across the five speech-and-audio benchmarks, either mean-HAG or variance-HAG attains the best scores in 13 of the 20 dataset × metric combinations, and at least one of the HAG algorithm performs best on 17 of the 20 dataset × metric combinations." I would translate this statement into a more structured experiment. One option is to rank the model giving 1 (best), 2 (second best), etc and some all the scores over the 20 combinations. Then plot the results to show that HAG variants reach an overall lower score (the lower the better).

(Remarks on code availability)

Reviewer #2

(Remarks to the Author)

The paper proposed the Hebbian Architecture Generation (HAG) method for reservoir computing, an unsupervised rule that grows connections between neurons that frequently activate together—embodying "fire together wire together" principle. Experiments were performed on a diverse set of classification and forecasting tasks which demonstrated the effectiveness of HAG over traditional Echo State Networks as well as reservoirs tuned with popular plasticity rules such as Intrinsic Plasticity or anti-Oja learning.

However, in terms a broader impact and the significance to the field of machine learning, the paper lacks a discussion on the comparison between recurrent neural networks and RC, where the former are much more popular for challenging classification and sequence forecasting tasks, such as LSTM, GRU, SSM and the Mamba architecture for LLM tasks. In a word, why use RC? what is the advantage of it compared to other alternatives?

(Remarks on code availability)

Version 1:

Reviewer comments:

Reviewer #1

(Remarks to the Author)

Authors addressed all my comments. I believe the paper is ready for publication.

(Remarks on code availability)

Reviewer #3

(Remarks to the Author)

This paper introduces a new approach to Reservoir Computing (RC) called Hebbian Architecture Generation (HAG). The method dynamically builds reservoir connections from an empty state, drawing inspiration from the biological principle that "neurons that fire together wire together". Unlike traditional Echo State Networks (ESNs) that rely on static, random architectures, HAG grows the connections in an unsupervised manner.

HAG-adapted reservoirs perform slightly better than traditional ESNs and other plasticity-based rules (Intrinsic Plasticity, Anti-Oja learning) across a diverse set of classification and forecasting tasks. In terms of efficiency, HAG's approach is computationally lightweight. Compared to expensive gradient-trained models like LSTMs and GRUs, particularly on small- and medium-sized datasets, the authors report that the network operates at a fraction of the computational cost. The authors also provide an analysis of how HAG reshapes the reservoir's internal dynamics. It effectively decorrelates neuron activities and expands the effective dimensionality of the state space, which directly leads to improved linear separability of data for classification tasks. This is a crucial finding, as it shows that HAG's unsupervised rule creates a more useful feature space for the downstream linear readout.

However, there is one significant conceptual point that needs clarification: the apparent contradiction between HAG's "expanded effective dimensionality" and its "low internal expressivity" compared to other methods. This nuance is critical and should be reconciled to strengthen the paper's central argument. A more precise explanation is needed to clarify that HAG's goal is to optimize for a task-relevant subspace rather than simply maximizing the number of principal components.

The methodology is generally sound and meets the expected standards for the field. The use of standard metrics, well-established baselines, and a transparent hyperparameter search protocol are all strong points. The paper includes a detailed computational complexity analysis and provides a principled reason for its design choices, such as using excitatory-only synapses.

A critical flaw in the data presentation is that while Tables 4 and 5 report the mean accuracy over eight trials, they lack

standard deviation values. Without this information, the mean value is not representative, as it fails to quantify the algorithm's variability and the reliability of the results.

Critical Review: Lack of Reproducibility

The most critical issue is the paper's lack of methodological precision, which severely impacts its reproducibility. Key parameters and procedures are vaguely defined or entirely absent, making it difficult for other researchers to replicate the results and verify the claims.

Missing Initialization Details: The paper mentions that input weights (W_{in}) are initialized from a normal distribution and a bias vector (b) is used, but it fails to specify the mean and standard deviation for the distribution, or how the bias is initialized. These are crucial parameters that can profoundly influence a reservoir's initial state and dynamics.

Vague Algorithmic Mechanisms: The process for random connection pruning and the method for sampling the dynamic time window ($T_{current}$) from a "log-space" are described without sufficient detail, introducing ambiguity into the core HAG algorithm.

Unquantified Variability: The paper acknowledges that the HAG algorithm's convergence is not guaranteed and that the final network configuration may vary across runs. However, it fails to quantify this variability by providing standard deviations for key metrics, which undermines the robustness of its claims and the reliability of the reported performance averages.

Conceptual Ambiguity and Limitations

The paper contains a significant conceptual contradiction that undermines one of its central arguments. It claims that HAG "expands the reservoir's 'effective' subspace" while simultaneously concluding that its "internal expressivity remains low compared to other methods". This inconsistency suggests a logical flaw in the analysis or a need for clearer terminology to explain HAG's unique operating point.

Furthermore, the paper's claims about HAG's efficacy are not universally supported across all tasks.

Subpar Forecasting Performance: The paper acknowledges that HAG's benefits are "less pronounced" and its performance more "variable" in time-series forecasting compared to classification. This suggests that HAG's correlation-driven, feature-decorrelating mechanism may disrupt the stable temporal representations needed for continuous prediction, a crucial function for many RC applications. This limitation is not adequately explored or conceptually explained.

Overly Broad Claims: The paper's conclusion that HAG provides a "practical and biologically plausible strategy" is a broad claim that lacks specific, quantitative support in certain contexts. The paper does not provide concrete examples of how HAG would excel in real-world scenarios beyond general "real-world time-series data".

Contextualizing Novelty:

While HAG's "from scratch" approach is presented as a novel contribution, the paper should more critically situate itself within the existing literature. The document itself references multiple prior attempts to introduce plasticity into RC models. A more critical review would acknowledge that HAG is not the first to apply Hebbian-inspired rules or seek to overcome the limitations of static RC. Instead, its novelty lies in the specific mechanism—using a longer-scale correlation to drive de novo structural growth—and the empirical evidence that this particular approach is more effective than prior methods, which focused on moment-to-moment updates or fine-tuning existing connections.

(Remarks on code availability)

The paper have only pseudocode.

Version 2:

Reviewer comments:

Reviewer #3

(Remarks to the Author)

Authors addressed all my comments, the paper is suitable for publication.

(Remarks on code availability)

The code appears to be reproducible, and I was able to follow it to assess the network described in the manuscript.

Answer for reviewer #1

Thank you for your insightful comments. We have addressed them as clearly as possible, both in the detailed answers below and in the revised version of our paper. We believe these changes have greatly improved the clarity and completeness of our manuscript.

In addition to the changes detailed below, we also added comparisons against GRU and LSTM baselines as suggested by Reviewer #2.

1. “To choose which connection to increase, for every neuron that has not yet achieved homeostasis, we compute pairwise linear correlation coefficients (Pearson, 1895) with every other neuron that is also not at homeostasis and we establish an incoming connection with the highest correlated neuron.” Provide an intuitive explanation for this methodological choice. Why do you think that increasing by Δw this particular connection yields the most benefits?

Response:

We now clarify that growth is driven by positive co-variation: for each under-active neuron we select the partner with the largest correlation (most often $r > 0$ as seen in Analysis) among neurons that are not at homeostasis, and we add an excitatory incoming edge from that partner. The benefit is to push units into the saturation region of tanh. You noted in question 4 that the link is symmetric, however, because neurons have different biases and inputs, the extra drive tends to redistribute their operating points in different regions of the tanh nonlinearity (one may sit closer to saturation while another remains nearer the quasi-linear zone). This creates heterogeneous instantaneous gains and curvatures across neurons over time, breaking synchrony and reducing redundancy. In mean-HAG this also raises the average activation toward the target band; in variance-HAG it increases diversity of local nonlinear responses.

Where to find the changes: Methods → HAG algorithm (§2.1): “Establishing the Connection” paragraphs; Fig. 2(b–d) (“HAG reshapes reservoir connectivity and dynamics”)

2. When decreasing a connection by Δw , the connection to prune is selected randomly. What is the rationale behind this choice?

Response:

We prune at random because we lack a reliable local signal for which removal increases global dimensionality; this avoids systematic bias (e.g., against recent edges) and breaks symmetries that would otherwise promote synchrony.

Where to find the changes: Methods → HAG algorithm (§2.1): end of paragraph starting “If $\Delta z_i > 1...$ ” explicitly stating random pruning and its rationale.v

3. Is the HAG process a converging method? What can we say about the existence and uniqueness of synaptic configurations that satisfy the HAG requirements? Does the HAG process always converge to the same configuration for a given input time series?

Response:

In practice, mean-HAG typically converges when provided with sufficient data windows, while variance-HAG may exhibit continuous rewiring in certain cases. The final configuration is generally not unique because of stochastic tie-breaks and random pruning. Given the complexity involving probabilistic adjustments, heterogeneous neuron plasticity schedules, and input dependency, a formal proof of convergence remains challenging and is part of ongoing research.

Where to find the changes: Methods → HAG algorithm (§2.1): Convergence paragraph (including the practical stabilization note and non-uniqueness).

4. Eq. 5. Due to symmetry, if the Pearson coefficient is maximised for (i,j) then it is maximised for (j*,i*). Then why one should increase by Δw only the connection (i*,j*) and not the connection (j*,i*)? Are those connections increased both? If so the Figure 2(b) is misleading.

Response:

We clarified explicitly in Section 2.1 that typically both connections are strengthened reciprocally within the same sweep, as identical correlation values for three or more neurons has a very low (though non-zero) probability. We revised Figure 2 to make this explicit, this figure is there so that the reader get a visual intuition that the network creates connection based on correlation which then disrupt synchronicity.

Where to find the changes: Methods → HAG algorithm (§2.1): paragraph beginning “Although updates are applied sequentially, in practice the resulting edge is reciprocal...”; Fig. 2 updated caption/panels.

5. How to select the desired homeostatic values of ρ_k and β_k ?

Response:

We explicitly state in Section 4.2 that ρ_k and β_k are hyperparameters optimized via the same Optuna/TPE search used for other parameters. Appendix A.2 lists their search ranges and final selected values.

6. Fig 5 describes a modular organisation of the input injected in the reservoir. How sensitive are the performance on this particular modularisation technique? Sometimes ESNs struggle to process high-dimensional inputs due to interference dynamics. Is the same modular injection applied for standard ESN?

Response:

We clarified in Section 2.3.4 that all ESN variants—static ESN, E-ESN, IP, Anti-Oja, IP+Anti-Oja, and both HAG variants—use the same modular input duplication; this ensures each input channel initially drives an equal number of neurons before recurrent mixing and is not a HAG-specific trick.

To address sensitivity, we added Appendix "Sensitivity to the Modularization of the Input Mapping", where we compare the modular mapping to an equivalently scaled/sparse random input mapping with hyperparameters tuned separately under each mapping and cross-evaluated. Across 15 dataset \times algorithm pairs, modularization yields a modest average gain of +1.03 pp; the effect is consistent on CatsDogs ($\approx 1-3$ pp for every rule) and small/mixed on Japanese Vowels and FSDD. The pooled direction is unchanged under cross-evaluation.

Where to find the changes: Appendix "Sensitivity to the Modularization of the Input Mapping" (new section with heatmap and summary).

7. The HAG method is claimed to be better than anti-Oja and Intrinsic Plasticity (IP), and this statement is supported by experiments using different metrics. However, the paper would benefit from a computational complexity analysis. Is the HAG computational time higher/lower than alternatives like IP or anti-Oja?

Response:

A new Section 2.5 and Table 2 and 3 were added, explicitly summarizing the asymptotic computational complexities of all evaluated models. HAG is shown to have comparable to Anti-Oja and orders of magnitude lower than LSTM /GRU which were added at the demand of both editor and reviewer2.

Where to find the changes: Methods \rightarrow Computational Complexity and Efficiency (new subsection) with Table 2 (asymptotics) and Table 3 with concrete FLOP estimates; Appendix "Computational Complexity" provides derivations for each method.

8. A Table 1. Maybe better Train Size and Test Size instead of Train Length and Test Length.

Response:

We updated terminology in Tables 1 and A.1 to replace “Train Length” and “Test Length” with “Train Size” and “Test Size” to enhance clarity. We could also use “Train steps”/“Test steps”.

Where to find the changes: Table 1 “Dataset details” now use “Train Size / Test Size”.

9. In pag 19 is reported “Across the five speech-and-audio benchmarks, either mean-HAG or variance-HAG attains the best scores in 13 of the 20 dataset × metric combinations, and at least one of the HAG algorithm performs best on 17 of the 20 dataset × metric combinations.” I would translate this statement into a more structured experiment. One option is to rank the model giving 1 (best), 2 (second best), etc and some all the scores over the 20 combinations. Then plot the results to show that HAG variants reach an overall lower score (the lower the better).

Response:

Implemented as Figure 6(e), we structured a comparative ranking across all dataset-metric combinations. This new figure clearly demonstrates that mean-HAG and variance-HAG consistently outperform static ESN and other plasticity models, providing robust support for our original statement.

Where to find the changes: Analysis → Separability and consistency of representations (§5.2): Fig. 6(e) “Cumulative rank sum across all clustering metrics” and accompanying paragraph reporting the cumulative rank results.

Thank you again for these thoughtful suggestions. We believe the revisions and new analyses address all points and improve the clarity and rigor of the work. We’re happy to provide any further detail if helpful.

Answer for reviewer #2

Thank you for your insightful comments. We have tried to address them as clearly as possible, both in the detailed answers below and in the revised version of our paper. We believe this greatly improved the clarity and completeness of our manuscript.

The paper proposed the Hebbian Architecture Generation (HAG) method for reservoir computing, an unsupervised rule that grows connections between neurons that frequently activate together—embodying "fire together wire together" principle. Experiments were performed on a diverse set of classification and forecasting tasks which demonstrated the effectiveness of HAG over traditional Echo State Networks as well as reservoirs tuned with popular plasticity rules such as Intrinsic Plasticity or anti-Oja learning. However, in terms a broader impact and the significance to the field of machine learning, the paper lacks a discussion on the comparison between recurrent neural networks and RC, where the former are much more popular for challenging classification and sequence forecasting tasks, such as LSTM, GRU, SSM and the Mamba architecture for LLM tasks. In a word, why use RC? what is the advantage of it compared to other alternatives?

Response:

We appreciate the request for broader context. We added a comparison to state of the art models, our comparisons emphasize recurrent models (ESN/HAG, LSTM, GRU) rather than SSMs (e.g., Mamba). This choice was made because all update a hidden state step-by-step; LSTM/GRU are therefore the most relevant gradient-trained baselines for RC. SSMs implement structured linear-time propagation with different computational primitives and objectives, so we treat them as orthogonal and leave them for future work.

Accordingly, we added LSTM and GRU baselines trained end-to-end on the same inputs and splits. Across tasks, HAG-ESN matches or exceeds LSTM/GRU on small/medium-data regimes (e.g., Japanese Vowels, CatsDogs, FSDD; and Mackey-Glass in forecasting), while on larger corpora (e.g., Spoken Arabic Digits, Speech Commands) LSTM/GRU lead as expected—but HAG remains competitive at far lower cost. Section 5 reports the accuracy summaries and a rank aggregation plot; Section 2.5 (with Table 2 and the OpCount table) details the compute gap, showing HAG's adaptation cost is comparable in order to Anti-Oja and orders of magnitude below LSTM/GRU. These findings align with prior work on RC's closed-form training, data efficiency [5-8], and low-power hardware realizations [1-4], while HAG adds label-free, online adaptation without sacrificing RC's simplicity.

Why RC (and where HAG fits):

1. Training efficiency. RC trains only a linear readout (closed-form ridge regression), avoiding BPTT; HAG preserves this—rewiring is unsupervised, label-

free, and the readout stays closed-form. Our complexity analysis (requested by Reviewer #1) shows orders-of-magnitude lower compute than LSTM/GRU for comparable hidden sizes. We also added a short positioning paragraph in Section 1 (“Static reservoir computing: strengths and limitations”).

2. Hardware & deployment. RC maps well to neuromorphic/photonic substrates with stringent energy/latency budgets [1–4]. HAG’s Hebbian excitatory-only weights are compatible with such platforms; we note this briefly in the HAG section and discuss further in the appendix.

On top of reservoir computing our HAG methods offer improved performances compared to previous unsupervised techniques, especially on bigger datasets, closing the gap with gradient based methods.

On top of GRU/LSTM, HAG remains better on datasets where samples are limited (a known limitation of gradient based RNN [5-8]) and offers unsupervised adaptation which make it possible to use HAG on partially unlabeled datasets. All this at a fraction of computational costs.

References

- [1] Yan, M. et al. 2024. Emerging opportunities and challenges for the future of reservoir computing. *Nature Communications* 15, 1. <https://doi.org/10.1038/s41467-024-45187-1>
- [2] Tanaka, G. et al. 2019. Recent advances in physical reservoir computing: A review. *Neural Networks* 115: 100-123. <https://doi.org/10.1016/j.neunet.2019.03.005>
- [3] Sozos, K. et al. High-speed photonic neuromorphic computing using recurrent optical spectrum slicing neural networks. *Communications Engineering* 1, 24 (2022). <https://doi.org/10.1038/s44172-022-00024-5>
- [4] Dongliang Wang, Yikun Nie, Gaolei Hu, Hon Ki Tsang, and Chaoran Huang. 2024. Ultrafast silicon photonic reservoir computing engine delivering over 200 TOPS. *Nature Communications* 15, 1. <https://doi.org/10.1038/s41467-024-55172-3>
- [5] Alwosheel, A., van Cranenburgh, S. & Chorus, C. 2018. Is your dataset big enough? Sample size requirements when using artificial neural networks for discrete choice analysis. *Journal of Choice Modelling* 28: 167-82. <https://doi.org/10.1016/j.jocm.2018.07.002>
- [6] Kavan Fatehi, Mercedes Torres Torres, and Ayse Kucukyilmaz. 2025. An overview of high-resource automatic speech recognition methods and their empirical evaluation in low-resource environments. *Speech Communication* 167: 103151. <https://doi.org/10.1016/j.specom.2024.103151>
- [7] Shewalkar, A., Nyavanandi, D. & Ludwig, S. 2019. Performance Evaluation of Deep Neural Networks Applied to Speech Recognition: RNN, LSTM and GRU. *Journal of Artificial Intelligence and Soft Computing Research* 9, 4: 235-245. <https://doi.org/10.2478/jaiscr-2019-0006>
- [8] Chattopadhyay, A., Hassanzadeh, P., Subramanian, D. & Fernando, B. 2019. Data-driven prediction of a multi-scale Lorenz 96 chaotic system using deep learning methods: Reservoir computing, ANN, and RNN-LSTM. <https://doi.org/10.48550/ARXIV.1906.08829>

Thank you again for prompting this broader context. We believe the new analyses and discussion address the request and clarify when RC (and HAG in particular) offers the best accuracy-efficiency trade-off. We are happy to elaborate further if helpful.

Answer for reviewer #3

Thank you for your insightful comments. We have addressed them as clearly as possible, both in the detailed answers below and in the revised version of our paper. We believe these changes have greatly improved the clarity and reproducibility of our work.

1. There is one significant conceptual point that needs clarification: the apparent contradiction between HAG's "expanded effective dimensionality" and its "low internal expressivity" compared to other methods. This nuance is critical and should be reconciled to strengthen the paper's central argument. A more precise explanation is needed to clarify that HAG's goal is to optimize for a task-relevant subspace rather than simply maximizing the number of principal components.

Response.

We agree the original phrasing was too loose. We now explicitly state that HAG expands the task-relevant subspace enough to improve linear separability. In short, HAG optimizes separability rather than indiscriminately maximizing the number of principal components, compared to fully signed reservoirs, whose CEVD can be larger yet not necessarily useful for the downstream task.

Where to find the changes:

- *Contributions* → *third entry updated to clarify HAG's goal*
- *Results* → *Conclusions* → *last paragraph*

2. A critical flaw in the data presentation is that while Tables 4 and 5 report the mean accuracy over eight trials, they lack standard deviation values. Without this information, the mean value is not representative, as it fails to quantify the algorithm's variability and the reliability of the results.

Response.

We now report mean \pm s.d. over 8 seeds for all test metrics in Tables 4 and 5. The other analysis metrics (which help quantify algorithm variability) are reported with dispersion in the Appendix.

Where to find the changes:

- *Results* → *Tables 4 and 5 now show mean \pm s.d.*

3. Missing Initialization Details: The paper mentions that input weights (W_{in}) are initialized from a normal distribution and a bias vector (b) is used, but it fails to specify the mean and standard deviation for the distribution, or how the bias

is initialized. These are crucial parameters that can profoundly influence a reservoir's initial state and dynamics.

Response. We now state the exact initialization details. We corrected one error which stated that W_{in} was drawn from a normal distribution while it's actually from uniform distribution. This is standard practice in reservoir computing (Lukoševičius (2012)). Biases follow a normal distribution with explicit mean and standard deviation.

Where to find the changes:

- *Methods* → *HAG algorithm (first paragraph)*. Input weights $W_{in} \sim \mathcal{U}(0,1)$ and biases $b_i \sim \mathcal{N}(0.1,0.1)$.

4. **Vague Algorithmic Mechanisms:** The process for random connection pruning and the method for sampling the dynamic time window ($T_{current}$) from a "log-space" are described without sufficient detail, introducing ambiguity into the core HAG algorithm.

Response. The core steps were under-specified (random pruning, log-spaced window sampling). We now fully specify both mechanisms in text, include the exact pruning equation, and have modified the pseudocode for completeness. The window length is sampled from a base-10 log-spaced grid between T_{min} and T_{max} ; pruning draws uniformly from the currently connected neurons and subtracts δw with non-negativity enforced.

Where to find the changes:

- *Methods* → *HAG algorithm: pruning equation and window selection clarified*.
- *Appendix A.1* → *"Pseudo-code for the HAG algorithm", which implements the log-spaced sampling and random pruning steps*.
- *Appendix B.1* → *Adaptive time window selection text*.

5. **Unquantified Variability:** The paper acknowledges that the HAG algorithm's convergence is not guaranteed and that the final network configuration may vary across runs. However, it fails to quantify this variability by providing standard deviations for key metrics, which undermines the robustness of its claims and the reliability of the reported performance averages.

Response. We added the standard deviations for accuracy and NRMSE. Analysis metrics are reported with variability in the Appendix.

Where to find the changes:

- *Appendix E* → *Detailed Analysis Results: dispersion for all analysis metrics.*
 - *Results: Tables 4 and 5 now report mean \pm s.d.*
-

6. The paper contains a significant conceptual contradiction that undermines one of its central arguments. It claims that HAG "expands the reservoir's 'effective' subspace" while simultaneously concluding that its "internal expressivity remains low compared to other methods". This inconsistency suggests a logical flaw in the analysis or a need for clearer terminology to explain HAG's unique operating point.

Response. This is now addressed by the clarifications in Item 1.

Where to find the changes:

- *Contributions* → *third entry updated to clarify HAG's goal*
 - *Results* → *Conclusions* → *last paragraph*
-

7. Subpar Forecasting Performance: The paper acknowledges that HAG's benefits are "less pronounced" and its performance more "variable" in time-series forecasting compared to classification. This suggests that HAG's correlation-driven, feature-decorrelating mechanism may disrupt the stable temporal representations needed for continuous prediction, a crucial function for many RC applications. This limitation is not adequately explored or conceptually explained.

Response. HAG's gains are smaller on forecasting tasks than on classification. HAG is designed to reduce redundancy and emphasize features that aid class separation, which is well suited to discrete classification problems but less directly beneficial for forecasting chaotic or quasi-periodic series (e.g., Mackey–Glass, Lorenz, Sunspots). These tasks reward maintaining stable, memory-bearing modes and long-range dependencies in addition to decorrelation.

That said, the improvements are "less pronounced" because HAG does not top individual datasets but *HAG does not disrupt temporal structure*: across forecasting benchmarks, mean-HAG attains the *best cumulative rank*, and variance-HAG performs *on par* with a standard ESN. Performance is also not more variable: the standard deviations are of the same order as those of other ESN variants and often lower than gradient-based techniques. We have therefore provided a more precise description in the Discussions.

Where to find the changes:

- *Results* → *standard deviations added in the NRMSE table (Table 5)*
- *Results* → *forecasting paragraph explicitly notes that HAG is above or comparable to other ESNs.*

- *Discussions and Implications for Reservoir Design* → second paragraph (reworded)
-

8. **Overly Broad Claims:** The paper's conclusion that HAG provides a "practical and biologically plausible strategy" is a broad claim that lacks specific, quantitative support in certain contexts. The paper does not provide concrete examples of how HAG would excel in real-world scenarios beyond general "real-world time-series data".

Response. We agree that the original statement was too broad. We have therefore narrowed and grounded the claim to the empirically demonstrated regimes presented in the paper. We also emphasize that HAG's practicality stems from its simplicity and compatibility with neuro-inspired or analog hardware, as outlined in Appendix "Differential Realization of Signed Weights for Positive-Only Reservoirs." While this alignment supports potential hardware implementations, it is not a central focus of the present work and is left to future research. We have also replaced the term "biologically plausible" with the more accurate "biologically motivated."

Where to find the changes:

- *Discussion* → concluding paragraphs (revised wording)
-

9. While HAG's "from scratch" approach is presented as a novel contribution, the paper should more critically situate itself within the existing literature. The document itself references multiple prior attempts to introduce plasticity into RC models. A more critical review would acknowledge that HAG is not the first to apply Hebbian-inspired rules or seek to overcome the limitations of static RC. Instead, its novelty lies in the specific mechanism—using a longer-scale correlation to drive de novo structural growth—and the empirical evidence that this particular approach is more effective than prior methods, which focused on moment-to-moment updates or fine-tuning existing connections.

Response. We agree that HAG is not the first to bring Hebbian ideas to RC. We synthesize prior work across 17 plasticity papers (15 Hebbian-type). To address the issue about scope, we clarified in the manuscript that HAG's contribution is this specific mechanism and its cross-benchmark performance, rather than the first use of Hebbian plasticity in RC.

Where to find the changes:

- *Contributions* → point "Longer-scale correlation as the driver" (comparison to previous methods more explicit)

10. The paper have only pseudocode.

Response. Apologies for the access issue. The Code Ocean capsule was inadvertently not publicly accessible because of a misconfiguration of the publication settings. This has been resolved: Code Ocean has verified the capsule and provided the editors with a peer-review copy. Upon acceptance, the capsule will be published; the permanent link is listed in the Code Availability section. The capsule contains all scripts and configuration needed to reproduce the tables and figures.

Where to find the changes:

▸ *Code availability: The link to the Code Ocean capsule (MIT license) contains all scripts to reproduce tables and figures.*

Thank you again for these thoughtful suggestions. We believe the revisions address all points and improve the clarity and rigor of the work. We're happy to provide any further details if helpful.